# n-3 Polyunsaturated Fatty Acids Are Associated with Stable Nitric Oxide Metabolites in Highly Trained Athletes

**DOI:** 10.3390/cells13131110

**Published:** 2024-06-27

**Authors:** Aleksandra Y. Lyudinina, Olga I. Parshukova, Evgeny R. Bojko

**Affiliations:** Institute of Physiology, Komi Science Centre, Ural Branch of the Russian Academy of Sciences, FRC Komi SC UB RAS, 50 Pervomayskaya Str., 167982 Syktyvkar, Russia; salu_06@inbox.ru (A.Y.L.); olga-parshukova@mail.ru (O.I.P.)

**Keywords:** n-3 polyunsaturated fatty acids, nitric oxide, nitrate, nitrite, cross-country skiers

## Abstract

Background: The aim of this study was to investigate the relationships between levels of n-3 essential polyunsaturated fatty acids (n-3 PUFAs) and stable nitric oxide (NO) metabolites in the plasma of athletes. Methods: Highly trained cross-country skiers (males, *n* = 39) were examined. The fatty acid profile of the total plasma lipids was determined by gas chromatography. The plasma NO level was studied by a colorimetric method via reaction with Griess reagent. Results: A widespread deficiency of essential n-3 PUFAs in the plasma of athletes (more than 80% of the subjects) was demonstrated in association with an imbalance in the levels of nitrates (NO_3_) and nitrites (NO_2_). A lower value of n-3 linolenic acid in the plasma (0.21 mol/%) was associated with a NO_3_ level below the normal range (n-3 C18:3 and NO_3_ Rs = 0.461; *p* = 0.003). Higher levels of n-3 eicosapentaenoic acid (0.8 mol/%) were associated with a concentration of NO_2_ above the normal value (n-3 C20:5 and NO_2_ Rs = 0.449; *p* = 0.004). Conclusion: For the first time, the participation of essential n-3 PUFAs in the nitrite–nitrate pathway of NO synthesis in highly trained skiers was demonstrated.

## 1. Introduction

A connection between endothelial function and fatty acid metabolism has been shown [1]. A special role in the functional state of the vascular endothelium is played by n-3 polyunsaturated fatty acids (n-3 PUFAs). Their positive effect on the functional state of blood vessels in healthy people has been proven [2,3], as has their ability to optimize physical performance [4]. Specifically, n-3 PUFAs are known to improve vascular endothelial function by increasing the formation and bioavailability of the endothelial-dependent relaxation factor nitric oxide (NO) through the activation of endothelial NO-synthase [5]. Supplementation with n-3 PUFAs increases vascular reactivity and the production of endogenous antioxidant enzymes and decreases the levels of inflammatory factors (such as interleukin-6 and tumor necrosis factor) [6,7]. In this regard, the prospects for studying the mechanisms that regulate the functional state of the vascular endothelium in the case of n-3 PUFA-enriched diets in athletes are highly relevant.

Several possible effects of n-3 PUFAs on NO synthesis have been discussed; however, the exact mechanisms of action of fatty acids have not been described. Thus, n-3 PUFAs are thought to contribute to increased NO availability [3,4,7,8]. Another pathway of NO synthesis has been associated with increased endothelial NO synthesis and even increased endothelial NO synthase activity [9,10]. Overall, these beneficial effects of n-3 PUFAs on NO levels may help to reduce systemic vascular resistance and blood pressure [11].

Cross-country skiers have a very high maximal oxygen uptake, and they are able to perform submaximal exercise at a rather high metabolic rate [4], which may lead to an intensification of the processes of production of reactive oxygen species. Thus, this group of athletes is a successful model for studying metabolic effects in humans during exercise, especially since trained cross-country skiers can actually perform difference-intensive work, revealing subtle regulatory mechanisms.

There have been several studies that have indicated the effects of n-3 PUFA intake on NO metabolite concentration and the functional state of the vascular endothelium among highly trained athletes [4,12]. Moreover, there are no data in the available literature on the combined relationship of these metabolites in the plasma with physical performance (PP) in highly trained athletes. Therefore, the purpose of this study was to investigate the relationships between levels of n-3 essential PUFAs and stable NO metabolites in the plasma of highly trained athletes.

## 2. Materials and Methods

### 2.1. Participants

The observation group included 39 highly trained cross-country skiers (males, aged 18–27 years), who were current members of the Komi Republic and national cross-country skiing team during the general training season. The anthropometric and physiological parameters of the examined athletes are presented in Table 1.

The group included 32 nonelite athletes who occupied the last ten places at official competitions and 7 elite athletes who occupied the first ten places at official competitions who were current members of the national team of Russia. All of the participants had more than 5 years of cross-country skiing practice as part of their main training schedule and had extensive experience in endurance events.

The criteria for qualifying for the study included age (18 to 30 years) and the availability of a medical certificate to confirm the athlete’s ability to practice sports (none of the athletes had bronchial asthma). Exclusion criteria included drug use, alcohol consumption greater than 1–2 drinks/week, a history of acute and exacerbated chronic illness, and signs of acute viral respiratory infection. All of the skiers were under the supervision of strength coaches and were following sport-specific training regimens unique to their respective sport.

### 2.2. Ethical Approval

The study conformed to the Code of Ethics of the World Medical Association (Declaration of Helsinki) (approval date: 1 November 2013, 28 December 2022 without No.). The volunteers were made aware of all of the information about the experimental protocol, experimental procedures, and probable risks and inconveniences associated with the performance of the exercise test on a cycle ergometer “until exhaustion”. After the necessary interpretations, the volunteers provided written informed consent to participate in the test. Participants were aware that they were free to leave the study at any time and without consequence. All of the skiers were supervised by a coach and followed athletic training regimens specific to their respective sport.

### 2.3. Study Procedure

The study was performed during the morning on an empty stomach; The low-nitrate dinner excluded meat and fish products, vegetables (mainly beets, leafy green vegetables), marinades, spirits, fruit, and other mineral drinks).

The height, body weight, and body composition of the athletes were measured by using a medical weight growth meter (Accuniq, SELVAS Health care, Daejeon, Republic of Korea). Body mass index (BMI) was calculated by using Quetelet’s formula. Systolic (SBP) and diastolic blood pressure (DBP) were measured by the Korotkov method using a Microlife Model BP AG1-30 device (Widnau, Switzerland).

### 2.4. Plasma Fatty Acid Profile Analysis

Samples of venous blood were collected from the skiers after at least 12 h of fasting. The plasma samples were collected into a vacutainer (Bekton Dickinson BP, London, UK) containing heparin as an anticoagulant. The plasma levels of polyunsaturated fatty acids (PUFAs) were determined by using gas-liquid chromatography. Sample preparation included extraction of the total lipids (fraction includes non-esterified FAs, phospholipids, triglycerides, and esterified cholesterol) from plasma and obtaining FA methyl ethers by using methanol and acetyl chloride, as described by the source [13]. Gas chromatography analysis of the PUFA methyl ethers was performed on a gas chromatograph («Crystal 2000M» Chromatek, Yoshkar-Ola, Russia) with a flame ionization detector attached to a SUPELCOWAX (25 m × 0.23 mm) capillary column (Supelco, St. Louis, MO, USA) at a temperature range of 170 °C to 250 °C (retention time—2 min). The temperature increase rate was 4 °C/min (overall time 25 min). Helium was used as the carrier gas, the volume rate was 0.6 mL/min, and the split was 1/65. The injector temperature was 260 °C, and the detector temperature was 200 °C. PUFA identification was performed by using Sigma standards. The quantitative analysis of the FAs concentrations was performed by using the internal standard of margaric (heptadecanoic) acid (C17:0). The PUFA concentrations are expressed as a weight percentage of the total weight of the fatty acids. The reported recommended fasting means of plasma total PUFAs (mol%) in healthy adults were taken from a preexisting source. We measured long-chain essential α-linolenic acid (C18:3 n-3, α-LNA), eicosapentaenoic acid (C20:5 n-3, EPA), and docosahexaenoic acid (C22:6 n-3, DHA) in total plasma lipids because they are significant biomarkers of short-term dietary intake of essential PUFAs [14].

### 2.5. Determination of NOx

The NO levels in the plasma were measured by using the Griess reaction indirectly by evaluating the stable metabolites of NO, including nitrites (NO_2_) and nitrates (NO_3_), the sum of which gives an indicator of the sum of stable NO metabolites (NOx). As previously described, nitrite and nitrate are the final products of NO in human plasma [15]. A strong correlation between endogenous NO production and NOx levels is known to exist in plasma [16]. After the deproteinization of the plasma samples in ethanol and centrifugation, the supernatants were incubated for 30 min at 37 °C with vanadium chloride to convert nitrate to nitrite. Afterward, the samples were mixed with Griess reagent. Samples were determined at a wavelength of 540 nm by using a Spectronic Genesys-6 Spectrophotometer (Thermo Electron Scientific Instruments LLC, Madison, WI, USA). Total nitrite was measured by using the Griess reagent. Samples were determined twice against a standard nitrite curve with a known concentration. The plasma nitrate level was calculated by subtracting the primary nitrite concentration from the total nitrite one. To determine the level of NO, we used a Sigma kit (St. Louis, MO, USA). The limit of detection NO was 0.001 mol/L. The NO_3_/NO_2_ index was calculated as the ratio between NO_3_ and NO_2_.

To examine the influence of n-3 PUFA levels on NOx, we divided the athletes into four groups relative to the reference norms of stable metabolites of nitric oxide [16]. Group I included athletes with a plasma NO_2_ level within the normal limits (*n* = 16), Group II included athletes with a plasma nitrite level that exceeded the normal limits (0–5 µmol/L) (*n* = 23), Group III included athletes with a level of NO_3_ in the plasma that was within the normal limits (*n* = 19), and Group IV included athletes with plasma nitrate levels that were below the reference limits (12–25 µmol/L) (*n* = 20).

### 2.6. Exercise Test on a Cycle Ergometer “Until Exhaustion”

On an ergometer bike (“Ergose-lect-100”, Ergoline GmbH, Hoechberg, Germany), physical and aerobic capacity (VO2max) tests were performed in breath-by-breath mode. The characteristics were averaged over 15 s segments. The test included one minute of cycling without load (for adaptation of the participants) followed by stepwise load increases of 40 W in 2 min increments. The first load started at 120 W. During the test, the pedaling speed was 60 rpm. Heart rate (HR) and workload were continuously recorded; additionally, measurements of VO2 and VCO2 were taken throughout the exercise by using an automated gas analysis system (Jaeger Oxycon Pro, Wuerzberg, Germany). The average VO2max was 59.7 ± 6.9 mL/min/kg.

### 2.7. Statistical Analysis

We performed the statistical procedures with Statistica software (version 12.6, StatSoft Inc., 2015, Tulsa, OK, USA). The statistical results are expressed as the mean ± standard deviation (M ± SD). We verified data normality with the Shapiro–Wilk test. We assessed the significance of differences between the indicators using the nonparametric Mann–Whitney U-test. A Spearman correlation test was used to analyze the correlation of stable metabolites of nitric oxide with α-LNA, EPA, and DHA. A value of *p* < 0.05 was accepted as statistically significant.

## 3. Results

During the preparation period, the FA profile of the plasma of the athletes was characterized by a deficiency of n-3 α-LNA (C18:3), EPA (C20:5), and DHA (C22:6) compared to the recommended standards (norms). The average level of essential α-LNA in the plasma of cross-country skiers was 0.3%, and the average levels of EPA and DHA of athletes were 0.7% and 1.4%, respectively, which were almost two times lower than the recommended limits [14]. A low proportion of α-LNA was noted in almost all of the skiers (EPA in 80% of athletes, and DHA in 94% of subjects), which is likely due not only to insufficient dietary intake of essential acids but also to the peculiarities of the metabolism of these FAs during intense physical activity (Table 2).

The level of NOx in the plasma of the examined athletes during the preparation period corresponded to the reference range [16] (Table 2). Moreover, the levels of stable NO metabolites (NO_2_ and NO_3_) were not included in the reference values. Thus, 64% of the athletes had an increased level of NO_2_ relative to the normal value, which averaged 10.2 µmol/L. In contrast, in half of the examined cross-country skiers, the level of NO_3_ was below the normal range (1.9–30.8 µmol/L).

Correlations between the plasma metabolites of the cross-country skiers are presented in Table 3.

A lower α-LNA value (0.21 mol/%) was associated with a lower NO_3_ concentration. The correlation coefficient between n-3 C18:3 and NO_3_ was 0.461 (*p* = 0.003) (Table 3). A higher EPA concentration (0.8 mol/%) was associated with higher NO_2_ levels. The correlation coefficient between n-3 C20:5 and NO_2_ was 0.449 (*p* = 0.004). Additionally, there was an inverse correlation between n-3 C20:5 and NO_3_ (−0.315) (*p* = 0.050).

Correlation analysis of the PP indicators demonstrated associations between FAs and the physical performance indicators, including between n-3 C18:3 and VO2max/kg (Rs = 0.486; *p* = 0.016), between C22:6 and VO2max (Rs = 0.406; *p* = 0.048), and between C22:6 and the HR baseline (Rs = 0.413; *p* = 0.045). A negative correlation was observed between the NO_2_ concentration and heart rate (Rs = −0.445; *p* = 0.048).

The results of the relationship between the level of essential PUFAs and the NOx indicator are presented in Table 4.

The average α-LNA value of approximately 0.4 mol/% corresponded to the generally accepted level of NO_3_ in the plasma of athletes. Furthermore, with pronounced α-LNA deficiency, significantly low levels of NO_3_ (*p* = 0.012) were detected in the plasma of skiers (Figure 1).

The average plasma EPA value of approximately 0.6 mol/% corresponded to the normal level of NO_2_ in the plasma. Higher EPA concentrations were detected with higher levels of NO_2_ (*p* = 0.026) in the plasma of skiers (Figure 2).

## 4. Discussion

In general, the potential mechanisms by which n-3 PUFAs improve vascular endothelial function and the cardiovascular system are currently under discussion. Mozaffarian D., Wu J.H. (2011), and other authors [9,11] noted that n-3 PUFAs affect many molecular pathways, including by eliciting changes in the physicochemical properties of cell membranes, direct interaction and modulation (or inhibition) of membrane ion channels and calcium regulatory proteins, regulation of gene expression with the help of nuclear receptors and transcription factors, changes in the eicosanoid profile, and the promotion of immunomodulatory and anti-inflammatory effects on the body [7,8,9,17].

Recent in vitro data suggest that transformations in membrane structure alter the functions of resident proteins, thus affecting cellular responses [18]. On the surface of mature endothelial cells, plasmalemma goblet-shaped depressions (also known as caveolae), are found, which play important roles in the compartmentalization, modulation, and integration of cellular signaling. Endothelial NO synthase is concentrated in caveolae and is in an inactive state in association with caveolin-1 and calmodulin [18]. Caveolae, in addition to PUFAs, contain a large number of receptors for proteins, acetylcholine, bradykinin, estrogens, and vascular endothelial growth factor; moreover, they play a decisive role in the regulation of endothelial vesicular transport [19]. For example, endothelium-independent vasodilation of the brachial artery is associated with membrane erythrocyte n-3 PUFA content in patients with arterial hypertension [20]. Dietary n-3 PUFAs may be involved in modulating the lipid composition of the apex of lipid rafts/caveolae microdomains, thereby influencing cell signaling, protein transport, and cellular cytokinetics [21,22], which subsequently affects the functional cell response. There is evidence of differential associations of EPA and DHA with a small number of caveolae/rafts associated with the protein Annexin A2 from the family of cytosolic Ca^2+^-binding proteins, which plays a key role in vascular function [23].

It is known that long-term and intense training (in conditions of oxygen deficiency), which is inherent in cross-country skiers, primarily affects the nitrite–nitrate (nonenzymatic) pathway of NO generation. Moreover, intense physical exercise weakens endothelial-dependent vasodilation, reduces the activity of endothelial NO synthase [24], and decreases the accumulation of NO [25,26] (Figure 3).

A previous study of endurance athletes demonstrated increased serum NO levels and overall optimization of physical performance after 3 weeks of n-3 PUFA supplementation [4]. Another study demonstrated that daily n-3 PUFA supplementation (0.87 g/day) for 3 weeks reduced airway inflammation in athletes [12]. The consumption of n-3 PUFAs leads to an increase in NO, reduces the vasoconstrictor response to norepinephrine and angiotensin 2, and enhances vasodilatory responses [11].

During the course of this study, we showed that the NOx indicator in the blood of the examined athletes during the preparation period corresponded to the literature data and reference values [16]. As we previously noted [27], in highly trained skiers during the competitive period, compared to the preparation period of the macrocycle, there was a decrease in the level of NO (mainly due to NO_3_), which can be observed in this study. A lower level of NO is accompanied by a deterioration in the adaptive capabilities of the athlete’s body due to a decrease in the enzymatic synthesis of NO, which can generally lead to a decrease in physical performance. An analysis of the PP indicators demonstrated associations between FAs and the physical performance indicators, including between n-3 C18:3 and VO2max/kg (Rs = 0.486; *p* = 0.016), between C22:6 and VO2max (Rs = 0.406; *p* = 0.048), and between C22:6 and the HR baseline (Rs = 0.413; *p* = 0.045), which is known from the source [4]. Due to the metabolic transformations of NO, NO_2_, and NO_3_, the optimal concentration of NO is maintained, which is necessary for the normal functioning of the cardiovascular system during exercise. We observed a negative correlation between the NO_2_ concentration and heart rate (Rs = −0.445; *p* = 0.048).

During the preparation period of training, according to our results, the NOx indicator of the athletes’ blood plasma corresponded to generally accepted reference values (22.0 ± 5.6 µmol/L). However, the NO_2_ levels were greater than normal, and the NO_3_ levels were within the lower limits of normal. When considering the direct correlation between n-3 C18:3 and NO_3_ (Rs = 0.461; *p* = 0.003), we can assume the indirect participation of α-LNA in maintaining the level of NO_3_ in the plasma blood within normal limits. This is also evidenced by the lower level of NO_3_ (*p* = 0.012) in the plasma of skiers with a more pronounced α-LNA deficiency relative to the normal range. Previously, we demonstrated a significant correlation between VO2max and NOx during maximum exercise in highly qualified athletes [15].

NO_3_ can be sequentially reduced to NO_2_ and NO, which correspondingly induces mitochondrial biogenesis in skeletal muscle cells through the activation of peroxisome proliferator-activated receptor-γ coactivator 1α (PGC-1α) [28]. The consumption of beetroot juice rich in NO_3_ may indeed be an ergogenic factor that promotes performance by doubling plasma NO_2_ levels and increasing exercise tolerance by 16% [10]. Additionally, the consumption of NO_3_ counteracts the hypoxic effect of PGC-1α (which plays a key role in the regulation of energy metabolism) in skeletal muscle, increases the activity of the nuclear receptor peroxisome proliferator-activated (PPAR β/δ), and consequently increases FA oxidation (both under normoxia and hypoxia) [29].

NO_2_ has been shown to play a special role as an alternative source of vasoactive NO [25]. Moreover, excess NO is converted into NO_2_, thus preserving the NO depot, which protects cells from oxidative and nitrosative stress. NO_2_ is present in sufficient quantities in the blood and can be enzymatically reduced to NO by xanthine oxidoreductase under conditions of low pH and pO2 [25]. This study suggests that exercise increases the use of NO_2_ as a precursor to mediate vasodilation [30]. Plasma NO_2_ levels were identified as being an important correlate of exercise tolerance in healthy individuals [31].

We have previously shown that with adequate dietary intake of n-3 PUFAs, a background deficiency of n-3 PUFAs in plasma was found in the majority of cross-country skiers [13,32]. Thus, the percent of essential α-LNA in the blood plasma of highly trained skiers with adequate dietary intake is 2.2 times lower than that in the blood plasma of students (similar age and gender to athletes). A suboptimal fatty acids diet was accompanied by an disproportion in the PUFAs profile in the plasma in both groups. Athletes have lower levels of saturated myristic and palmitic acids, which are within the reference ranges. The level of EPA in the plasma in both groups was also reduced, and in students, it was almost 3 times more pronounced than in athletes. Thus, the consumption of n-3 PUFAs by skiers in accordance with the recommended value does not cover their energy intake and physiological functions involved in intense physical exercise, which reduces the aerobic performance of the body [32].

α-LNA is an important essential fatty acid, the main source of which, in addition to perilla oil (58%), is flaxseed oil (57%), as well as soybean and rapeseed oils (approximately 10%). Moreover, the significant role of α-LNA in the implementation of a widespread range of different functions (for example, its active participation in the energy supply of aerobic power of the body [13] and in the regulation of heart rhythm [33]) determines the need to compensate for the deficiency of its content in the body. The active participation of α-LNA in the metabolism of essential FAs has also been demonstrated by increasing the activity of FA desaturases in liver microsomes, which correspondingly leads to the conversion of α-LNA into EPA, which is one of the main bioregulators of the body [14].

A mechanism has been described that demonstrates the effect of EPA on increasing NO bioavailability (including on NO synthesis and endothelial function) through the activation of AMP-activated protein kinase in response to intracellular ATP depletion [8,34,35]. According to our data, the average EPA value (approximately 0.6 mol/%) was below the recommended range (1.0–2.2 mol/%), while corresponding to the normal NO_2_ level in the plasma; moreover, with a higher EPA value, a higher NO_2_ level was indicated (*p* = 0.026) in the blood of skiers. Spearman’s correlation coefficient between n-3 C20:5 and NO_2_ was 0.449 (*p* = 0.004). A deficiency of EPA in the plasma of the subjects (0.6 ± 0.5 mol%) not only was related to dietary habits but also could indicate the neutralization of oxidative stress in the body because EPA inhibits the processes of lipid peroxidation in membrane vesicles [36]. EPA is beneficial for endothelial function; it improves the balance between NO and peroxynitrite and acts synergistically with statins [36]. In addition, EPA—through the regulation of long-chain acyl-CoA synthetase expression—reduces vascular endothelial dysfunction induced by palmitic acid-induced ROS formation [37]. Thus, oxidative stress is the main reason for the decrease in the activity of NO synthase through a decline in the availability of the cofactor NOS-tetrahydrobiopterin and, subsequently, the inhibition of the enzymatic synthesis of NO [24]. The mechanism by which EPA influences the development of atherosclerosis involves influencing endothelial dysfunction and oxidative stress and enhancing the synthesis of eicosanoids (which dilate blood vessels and reduce thrombus formation and inflammation), thus reducing the risk of atherogenic dyslipoproteinemia [38]. In addition, essential n-3 PUFAs not only enhance the formation of NO but also react with it to form the corresponding nitroalkene derivatives, which promote vascular dilation, inhibit neutrophil degranulation and superoxide formation, inhibit platelet activation, and increase PPAR activity and NO release. This mechanism prevents platelet aggregation, blood clot formation, atherosclerosis, and cardiovascular diseases [11,17,20]. Collectively, these effects may be the subject of future discussions about the benefits of n-3 PUFA-enriched diets in regulating vascular tone and increasing physical performance and endurance in general.

### Limitations

This study is not without limitations. First, the study sample is relatively small, so further studies with larger athlete populations are needed to define the precise effect of different classes of fatty acids (saturated and n-6 PUFAs) on endothelial function in skiers. Second, a control group was not included in the study design. Third, from a statistical standpoint, this study lacks the power to assess causality, and there is the risk of spurious correlations between variables because of multiple statistical tests. In addition, when working with athletes, there is always the possibility of individual differences and deviations.

## 5. Conclusions

For the first time, we demonstrated the association of essential n-3 PUFAs with the plasma level of NO and the possible participation of n-3 PUFAs in the nitrite–nitrate pathway of nitric oxide synthesis in highly trained skiers. It has been established that a higher value of essential eicosapentaenoic acid promotes the accumulation of nitrites (NO_2_) in the blood of athletes, thus creating conditions for the production/deposition of nitric oxide. Higher levels of essential α-LNA in the plasma promote an increase in nitrate (NO_3_) levels while also regulating nitric oxide (NO) stores, which are necessary to maintain vascular tone and generally normal functioning of the cardiovascular system in highly trained athletes.

## Figures and Tables

**Figure 1 cells-13-01110-f001:**
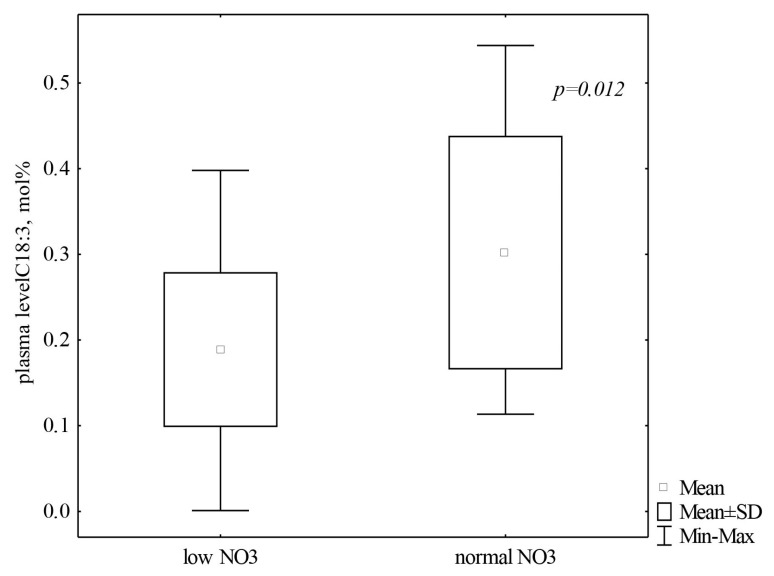
Distribution of α-linolenic acid in the plasma of athletes (*n* = 37) depending on the level of NO_3_. Note: Data are presented as M ± SD. C18:3—α-linolenic acid.

**Figure 2 cells-13-01110-f002:**
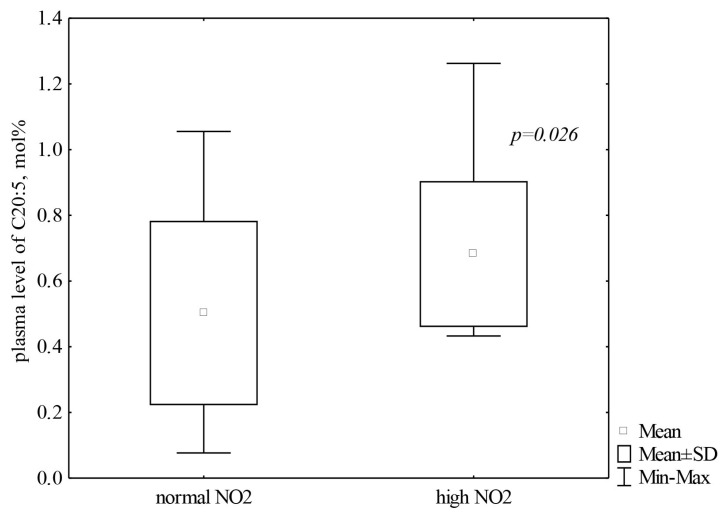
Distribution of eicosapentaenoic acid in the plasma of athletes (*n* = 37) depending on the level of NO_2_. Note: Data are presented as M ± SD. C20:5—eicosapentaenoic acid.

**Figure 3 cells-13-01110-f003:**
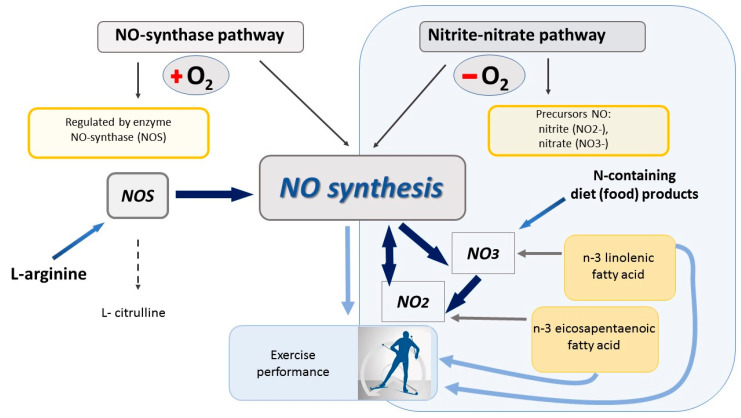
The main metabolic pathways for the synthesis of nitric oxide and the participation of essential fatty acids in the production of NO_2_ and NO_3_.

**Table 1 cells-13-01110-t001:** The anthropophysiometric characteristics of the examined cross-country skiers at baseline.

Characteristics of Athletes	M ± SD
Age, years	21.7 ± 3.9
Height, cm	176.1 ± 4.8
Body mass, kg	72.8 ± 4.1
Body-weight index, kg/m^2^	22.8 ± 1.1
Fat mass, %	9.6 ± 2.7
Maximum oxygen consumption, L/min	4.1 ± 0.6
Heart rate	60.4 ± 11.3
Systolic blood pressure, mm Hg	117.0 ± 11.2
Diastolic blood pressure, mm Hg	71.8 ± 8.4

**Table 2 cells-13-01110-t002:** The plasma metabolites of cross-country skiers relative to recommended norms.

Metabolites	M ± SD	Reference Range
α-Linolenic acid (C18:3 n-3, mol% (mg/mL)	0.3 ± 0.2 0.005 ± 0.002	>0.6 mol%
Eicosapentaenoic acid (C20:5 n-3, mol% (mg/mL)	0.7 ± 0.4 0.012 ± 0.008	>1.4 mol%
Docosahexaenoic acid (C22:6 n-3, mol% (mg/mL)	1.4 ± 0.7 0.023 ± 0.011	>2.4 mol%
NOx, µmol/L	22.0 ± 5.6	17–35
NO_2_, µmol/L	10.2 ± 5.2	0–5
NO_3_, µmol/L	11.8 ± 6.3	12–25

Note: Reference ranges for essential fatty acids in the plasma blood are taken from (Hodson et al., 2008) [14] reference ranges for stable nitric oxide metabolites are taken from (Granger et al., 1996) [16].

**Table 3 cells-13-01110-t003:** Correlation between n-3 PUFAs and stable NOx metabolites in the plasma of cross-country skiers.

Parameters	Spearman Correlation (Rs)	*p* Level
C18:3 n3 and NOx	0.255	0.115
C18:3 n3 and NO_2_	−0.304	0.059
C18:3 n3 and NO_3_	0.461	0.003
C20:5 and NOx	0.116	0.481
C20:5 and NO_2_	0.449	0.004
C20:5 and NO_3_	−0.315	0.050
C22:6 and NOx	0.025	0.877
C22:6 and NO_2_	−0.192	0.239
C22:6 and NO_3_	0.195	0.233

**Table 4 cells-13-01110-t004:** Level of essential n-3 PUFAs in comparison with stable NOx metabolites in the plasma of cross-country skiers.

	α-Linolenic Acid (C18:3 n-3, mol%)	Eicosapentaenoic Acid (C20:5 n-3, mol%)	Docosahexaenoic Acid (C22:6 n-3, mol%)
**Nitrites—NO_2_**			
I Group	0.37 ± 0.39	0.56 ± 0.37	1.47 ± 0.89
II Group	0.24 ± 0.09	0.79 ± 0.38	1.38 ± 0.60
*p*-level	0.242	0.0178	0.819
**Nitrates—NO_3_**			
III Group	0.4 ± 0.35	0.83 ± 0.53	1.51 ± 0.82
VI Group	0.21 ± 0.06	0.77 ± 0.25	0.91 ± 0.25
*p*-level	0.0169	0.168	0.354

## Data Availability

The original contributions presented in the study are included in the article, further inquiries can be directed to the corresponding author.

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
