# Peer review of "n-3 Polyunsaturated Fatty Acids Are Associated with Stable Nitric Oxide Metabolites in Highly Trained Athletes"

_cells, 2024, doi:10.3390/cells13131110_

Round 1

Reviewer 1 Report

Comments and Suggestions for Authors

The article “n-3 polyunsaturated fatty acids regulate the stable nitric oxide metabolites in highly trained athletes” aimed to profile the n-3 fatty acids and No levels in plasma samples obtained from athletes. The authors used GC for fatty acid determination and colorimetric method for NO determination. The authors found the association between C18:3 and C20:5 n-3 fatty acids with NO3 and NO2. Although the study is interesting however, I do have several concerns about the experiments conducted.

Introduction should be revised and there are several sentences are incomplete: as for example “N-3 polyunsaturated fatty acids (n-3 PUFAs) play a special role in this process” what process?

Table 1 what is M?  use pullstop instead of comma for numbers.

Ethical approval: provide the approval number.

Line 92: Gasliquid chromatography analysis? What does it mean

The authors used margarine acid (C17:0) as an internal standard but it is present endogenously which questions the reliability of quantification data. Deuterated internal standards are essential for such analysis. Moreover, for polyunsaturated fatty acids why the saturated fatty acid internal standard was selected?

Line 104: ALA, EPA, DHA should be expanded when using first time. Why n-6 fatty acids are determined in the study despite the authors mention that n-6/n-3 ratio is important factor for cardiovascular risk .

-Blood or plasma unify the terminology

-provide the section numbers in methods section.

-Table 2 data is not very convincing: provide the absolute concentration values along with mol%

- Table 3: it shows very poor correlation coefficients. For mentioning p-value what type of statical test were applied is missing.

- Figure 1 and 2: show all the individual values in the graph.
- Discussion should be more relevant with the authors results.

-α-LNA or ALA  unify the terminologies.

Comments on the Quality of English Language

Check with a native speaker.

Author Response

Dear reviewers and editor!

Thank you and Reviewers for a very extensive and detailed review of our manuscript. We have changed the manuscript based on the reviewers’ comments. Hopefully the manuscript can be published now. Our responses to the comments of the reviewer are presented below.

Sincerely, all authors (Aleksandra Lyudinina,  Olga Parshukova and Evgeny Bojko)

Reviewers 1

The article “n-3 polyunsaturated fatty acids regulate the stable nitric oxide metabolites in highly trained athletes” aimed to profile the n-3 fatty acids and No levels in plasma samples obtained from athletes. The authors used GC for fatty acid determination and colorimetric method for NO determination. The authors found the association between C18:3 and C20:5 n-3 fatty acids with NO3 and NO2. Although the study is interesting however, I do have several concerns about the experiments conducted.

- Introduction should be revised and there are several sentences are incomplete: as for example “N-3 polyunsaturated fatty acids (n-3 PUFAs) play a special role in this process” what process?

Thank you for your remark. We revised the introduction and added the phrase “A special role in the functional state of the vascular endothelium play n-3 polyunsaturated fatty acids (n-3 PUFAs)”

- Table 1 what is M? use pullstop instead of comma for numbers.

The results are expressed as the mean ± standard deviation (M±SD). Commas in all tables were replaced on the pullstop.

- Ethical approval: provide the approval number.

We added information in MM. “The study was conducted in accordance with the Declaration of Helsinki, and approved by the Local Research Bioethics Committee of the Institute of Physiology of the Komi Scientific Centre of the Ural Branch of the Russian Academy of Sciences (approval date: 1 November 2013, 28 December 2022 without No)”.

- Line 92: Gas‒liquid chromatography analysis? What does it mean

Gas-liquid chromatography is a type of gas chromatography.

- The authors used margarine acid (C17:0) as an internal standard but it is present endogenously which questions the reliability of quantification data. Deuterated internal standards are essential for such analysis. Moreover, for polyunsaturated fatty acids why the saturated fatty acid internal standard was selected?

Thank you for your valuable comment. In the future we will definitely use deuterated derivatives of fatty acids. Margaric acid is a minor compound of human lipids and is often used as an internal standard in the analysis of fatty acids. One of the links is attached (Abdelmagid SA, Clarke SE, Nielsen DE, Badawi A, El-Sohemy A, Mutch DM, et al. (2015) Comprehensive Profiling of Plasma Fatty Acid Concentrations in Young Healthy Canadian Adults. // PLoS ONE 10(2): e0116195. doi:10.1371/journal.pone.0116195).

- Line 104: ALA, EPA, DHA should be expanded when using first time. Why n-6 fatty acids are determined in the study despite the authors mention that n-6/n-3 ratio is important factor for cardiovascular risk.

We have taken account. The article did not examine the relationship between nitric oxide levels and n-6 PUFAs.

- Blood or plasma unify the terminology

We have made edits

- provide the section numbers in methods section.

We have taken account.

-Table 2 data is not very convincing: provide the absolute concentration values along with mol%

We added absolute concentration values of PUFAs in Table 2

- Table 3: it shows very poor correlation coefficients. For mentioning p-value what type of statical test were applied is missing.

A Spearman correlation test was used to analyse the correlation of stable metabolites of nitric oxide with α-LNA, EPA and DHA

- Figure 1 and 2: show all the individual values in the graph.

in figure 1 and 2 individual values were added and slightly adjusted with extreme outliers. Leave it to your own discretion to find the best version.

- Discussion should be more relevant with the authors results.

We hope our edits improved the conclusion.

-α-LNA or ALA unify the terminologies.

We have made edits (α-LNA throughout the text)

Reviewer 2 Report

Comments and Suggestions for Authors

The aim was to investigate the relationships between n-3 PUFAs and stable NO metabolites in the blood plasma of athletes. The plasma NO level was study by colorimetric method via reaction with the Griss reagent. The authors claim that: (1) Higher values of EPA promote accumulation of nitrites in the blood of athletes, thus creating conditions for the production of NO; (2) Higher levels of ALA promote an increase in nitrate levels while also regulating NO stores.

In general, the work is rather weak from a scientific and methodological point of view; in addition, it does not correspond to the profile of the journal. If we focus on MDPI journals, then Sports or Metabolites might be more suitable.

The authors attempted a systematic analysis of biochemical processes in the body of athletes based on a very limited set of biochemical and physiological data; the role of individual cells and tissues is determined very speculatively. Correlations at the level of 0.4-0.5 are very unreliable for the conclusions that the authors draw, but even a higher strength of the relationship between individual parameters does not give an unambiguous picture of cause-and-effect relationships. To do this, it is necessary to study dozens of other physiological and biochemical indicators, conduct other types of correlation and/or cluster analysis, and conduct a broader and deeper analysis of modern scientific literature.

Of the 37 references, only 6 are from the last 5 years, of which three are to the authors’ own works. But even referring to old classic works related to determining the relationship between the level of nitrites/nitrates and the generation of nitric oxide, the authors ignored many tips and warnings and did not fulfill important conditions for conducting such experiments.

For in vivo studies on NOS activity, experimental animals and human subjects must be maintained on a chemically defined nitrite/nitrate-free diet or a diet with a low nitrite/nitrate content (<180/umol/day for humans). In the intestine, reduction of nitrites and nitrates to N2 and NH3 by gut microbial flora takes place. Because this partitioning appears to be relatively constant under various physiological and pathophysiological conditions, measurement of daily urinary nitrate excretion in the face of zero or low nitrate intake, has proved to be a reliable and quantitative estimate of nitric oxide production in vivo. Circulating nitrite reflects constitutive endothelial NOS activity, whereas excretory nitrate indicates systemic NO production.

Despite the chemical simplicity of nitrite and nitrate, accurate and interference-free quantification of nitrite and nitrate in biological fluids as indicators of NO synthesis may be difficult. Nitric oxide can react with O2•− to form peroxynitrite (ONOO−), which then can yield nitrate (NO3) or be reduced to NO2 and NO2. NO3 can be also derived from the reaction between NO and oxyhemoglobin (HbO2). Nitrosyl (HNO) can be formed by reduction of NO. NO can bind to iron (Fe(II)) to form dinitrosyl iron complexes (DNICs). NO can also react with nitrogen dioxide (NO2), with N2O3 as intermediate, to generate nitrite (NO2) and nitrosated product (RXNO).

Also, nitrite produced in vivo reacts with oxyhemoglobin once it enters the vascular system. The reaction of nitrite with oxyhemoglobin results in stoichiometric formation of nitrate and methemoglobin. Thus NO synthesized from L-arginine is detected in plasma, serum, and urine as nitrate. However, nitrite that enters body fluids without traversing a vascular space may not be completely oxidized to nitrate. Significant amounts of nitrite could accumulate in cerebrospinal fluid, joint fluid, and exudative fluids in the pleural, pericardial, and peritoneal cavities.

The authors' interpretation of the data obtained is highly speculative and, as a rule, does not correspond to the conclusions of the papers cited by the authors, without additional experiments or without obtaining additional data in the current experiment. In addition, the article contains many typos or errors which indicate either a lack of understanding by the authors of the nuances of the English language, or a lack of understanding of the scientific essence of the phenomenon, or both. This is primarily the title of the article, as well as phrases like “activation of endothelial NO”, «anti-inflammatory factors (such as interleukin-6 and tumour necrosis factor)», «endothelial NOx synthesis», «Height, m  1,7±4,8», «essential n-3 ALA, EPA and DHA», «The NO3/NO2 ratio was calculated as the ratio between NO3 and NO2», «bioaccumulation of NO», «consumption of NO3 counteracts the hypoxic effect of PGC-1», «NO2 is present in sufficient quantities in the blood and can be enzymatically reduced to NO by xanthine oxidoreductase and nonenzymatically reduced under conditions of low pH and pO2», «deficiency of EPA in the blood of the subjects (0.6±0.5 mol%) may indicate active oxidative stress in the body because EPA inhibits the processes of lipid peroxidation in membrane vesicles, and EPA also improves the balance between NO and peroxynitrite and acts synergistically with statins».

Comments on the Quality of English Language

The article contains many typos or errors which indicate either a lack of understanding by the authors of the nuances of the English language, or a lack of understanding of the scientific essence of the phenomenon, or both. This is primarily the title of the article, as well as phrases like “activation of endothelial NO”, «anti-inflammatory factors (such as interleukin-6 and tumour necrosis factor)», «endothelial NOx synthesis», «Height, m  1,7±4,8», «essential n-3 ALA, EPA and DHA», «The NO3/NO2 ratio was calculated as the ratio between NO3 and NO2», «bioaccumulation of NO», «consumption of NO3 counteracts the hypoxic effect of PGC-1», «NO2 is present in sufficient quantities in the blood and can be enzymatically reduced to NO by xanthine oxidoreductase and nonenzymatically reduced under conditions of low pH and pO2», «deficiency of EPA in the blood of the subjects (0.6±0.5 mol%) may indicate active oxidative stress in the body because EPA inhibits the processes of lipid peroxidation in membrane vesicles, and EPA also improves the balance between NO and peroxynitrite and acts synergistically with statins».

Author Response

Dear reviewers and editor !

Thank you and Reviewers for a very extensive and detailed review of our manuscript. We have changed the manuscript based on the reviewers’ comments. Hopefully the manuscript can be published now. Our responses to the comments of the reviewer are presented below.

Sincerely, all authors (Aleksandra Lyudinina,  Olga Parshukova and Evgeny Bojko)

Reviewers 2

The aim was to investigate the relationships between n-3 PUFAs and stable NO metabolites in the blood plasma of athletes. The plasma NO level was study by colorimetric method via reaction with the Griss reagent. The authors claim that: (1) Higher values of EPA promote accumulation of nitrites in the blood of athletes, thus creating conditions for the production of NO; (2) Higher levels of ALA promote an increase in nitrate levels while also regulating NO stores.

In general, the work is rather weak from a scientific and methodological point of view; in addition, it does not correspond to the profile of the journal. If we focus on MDPI journals, then Sports or Metabolites might be more suitable.

The authors attempted a systematic analysis of biochemical processes in the body of athletes based on a very limited set of biochemical and physiological data; the role of individual cells and tissues is determined very speculatively. Correlations at the level of 0.4-0.5 are very unreliable for the conclusions that the authors draw, but even a higher strength of the relationship between individual parameters does not give an unambiguous picture of cause-and-effect relationships. To do this, it is necessary to study dozens of other physiological and biochemical indicators, conduct other types of correlation and/or cluster analysis, and conduct a broader and deeper analysis of modern scientific literature.

Of the 37 references, only 6 are from the last 5 years, of which three are to the authors’ own works. But even referring to old classic works related to determining the relationship between the level of nitrites/nitrates and the generation of nitric oxide, the authors ignored many tips and warnings and did not fulfill important conditions for conducting such experiments.

For in vivo studies on NOS activity, experimental animals and human subjects must be maintained on a chemically defined nitrite/nitrate-free diet or a diet with a low nitrite/nitrate content (<180/umol/day for humans). In the intestine, reduction of nitrites and nitrates to N2 and NH3 by gut microbial flora takes place. Because this partitioning appears to be relatively constant under various physiological and pathophysiological conditions, measurement of daily urinary nitrate excretion in the face of zero or low nitrate intake, has proved to be a reliable and quantitative estimate of nitric oxide production in vivo. Circulating nitrite reflects constitutive endothelial NOS activity, whereas excretory nitrate indicates systemic NO production.

Despite the chemical simplicity of nitrite and nitrate, accurate and interference-free quantification of nitrite and nitrate in biological fluids as indicators of NO synthesis may be difficult. Nitric oxide can react with O2•− to form peroxynitrite (ONOO−), which then can yield nitrate (NO3) or be reduced to NO2 and NO2. NO3 can be also derived from the reaction between NO and oxyhemoglobin (HbO2). Nitrosyl (HNO) can be formed by reduction of NO. NO can bind to iron (Fe(II)) to form dinitrosyl iron complexes (DNICs). NO can also react with nitrogen dioxide (NO2), with N2O3 as intermediate, to generate nitrite (NO2) and nitrosated product (RXNO).

Also, nitrite produced in vivo reacts with oxyhemoglobin once it enters the vascular system. The reaction of nitrite with oxyhemoglobin results in stoichiometric formation of nitrate and methemoglobin. Thus NO synthesized from L-arginine is detected in plasma, serum, and urine as nitrate. However, nitrite that enters body fluids without traversing a vascular space may not be completely oxidized to nitrate. Significant amounts of nitrite could accumulate in cerebrospinal fluid, joint fluid, and exudative fluids in the pleural, pericardial, and peritoneal cavities.

The authors' interpretation of the data obtained is highly speculative and, as a rule, does not correspond to the conclusions of the papers cited by the authors, without additional experiments or without obtaining additional data in the current experiment. In addition, the article contains many typos or errors which indicate either a lack of understanding by the authors of the nuances of the English language, or a lack of understanding of the scientific essence of the phenomenon, or both.

Esteemed Reviewer.

First of all, let me thank You for your deep and succinct review of our manuscript. We are grateful for information provided in Your review, with modern understandings of the metabolism of NO and its stable products in the body.

To explain our conclusions on the materials provided in our work we would like to note that many aspects of the NO metabolism demonstrated in literature were obtained on various small-scale – cellular or even sub-cellular – models, and not all of them were constructed using human cells [for example, Martins et al., 2014, Philip et al., 2024]. In our manuscript we assessed NO-associated processes in the whole human organism, and demonstrated a phenomenon previously not described in the literature.

The particular feature of our model is that the volunteer in our study were elite cross-country ski runners, current members of the national team, which is one of the two strongest world teams in this sport, and our participants lacked TUEs (therapeutic use exemptions) unlike the members of the Norway team. It is in this sport the highest maximal oxygen uptake values among the athletes were detected. Such high oxygen utilization ought to be followed, to a certain degree, by an increased generation of active oxygen species.

We believe that those facts make our proposed scientific contribution both new and significant, since they were obtained in this particular model in the test “to exhaustion”. We don’t doubt that our conclusion about an interconnection between the biochemical parameters (“For the first time, the participation of essential n-3PUFAs in the nitrite-nitrate pathway of nitric oxide synthesis in highly-trained skiers has been demonstrated”) truly demonstrates the nature of intersection of the metabolic pathways related to the free radical oxidation of fatty acid. The data we obtained contributes to the study of cellular processes. That was the reason for us to choose the “Cells” journal, and to submit our manuscript to the special issue “'Molecular Mechanisms of Exercise and Healthspan”. This is the first report in the literature using such type of the experimental model, and, most certainly, in our future work we will continue studying the complex mechanisms underlying this phenomenon, according to the recommendations of the esteemed Reviewer.

This is primarily the title of the article as well as phrases like “activation of endothelial NO”, «anti-inflammatory factors (such as interleukin-6 and tumour necrosis factor)», «endothelial NOx synthesis», «Height, m 1,7±4,8», «essential n-3 ALA, EPA and DHA», «The NO3/NO2 ratio was calculated as the ratio between NO3 and NO2», «bioaccumulation of NO», «consumption of NO3 counteracts the hypoxic effect of PGC-1», «NO2 is present in sufficient quantities in the blood and can be enzymatically reduced to NO by xanthine oxidoreductase and nonenzymatically reduced under conditions of low pH and pO2», «deficiency of EPA in the blood of the subjects (0.6±0.5 mol%) may indicate active oxidative stress in the body because EPA inhibits the processes of lipid peroxidation in membrane vesicles, and EPA also improves the balance between NO and peroxynitrite and acts synergistically with statins».

Thank you for your remark. We considered this expert comment.

Reviewer 3 Report

Comments and Suggestions for Authors

Article Cells-3038568:

 n-3 polyunsaturated fatty acids regulate the stable nitric oxide 2 metabolites in highly trained athletes

Comments for authors.

In this study, the authors investigated the relationships between levels of n-3 polyunsaturated essential fatty acids (n-3 PUFAs) and stable nitric oxide (NO) metabolites in the blood plasma of athletes. These are my suggestions for the manuscript.

The title is pretentious, so the word " regulate" in the title needs to be changed to "has a relationship with...." or "is associated with .......".

Methods: Please describe the method of transesterification of fatty acids with methanol and acetyl chloride. The reference you provided (#13) does not describe this method. It is important because we need to know if you methylated the fatty acids in the whole plasma lipids or only the non-esterified fatty acids (NEFA).

Line 98: The evaporator temperature was 260°C, ….did you mean the injector……

Line 100: …..margarine acid  solution (C17:0). Did you mean margaric acid (heptadecanoic acid)

Line 103: We have measured the essential n-3 103 ALA, EPA and DHA... Of these fatty acids, only ALA is essential, as it is not synthesized in the body and must be ingested with food. EPA and DHA, however, are synthesized in the body by the enzymatic system of desaturase and elongase and are not essential fatty acids.

Line 110: Please, precisely define NOx. From the sentence, it is not clear what ratio is NOx.

It is not clear why the Exercise Test on a Cycle Ergometer "until Exhaustion" was done because all the results presented in the paper refer only to baselines.

The statistics chapter is missing, so it is not clear how the results were processed.

As for the methods in general, they are written almost identically to the methods in their other papers related to skiers, some of which were also published in the Cells. In my opinion, it needs to be reformulated.

Results:

The abbreviations for FA must be marked the same throughout the work. Namely, in the first part of the paper, you marked alpha-linolenic acid as ALA, and then in the results with α-LNA. Introduce the abbreviations when you first mention these FAs, that is, in the methodology, stating the full name of fatty acids just as you put in Table 2, and the abbreviation for alpha-linolenic acid can be ALA because it is more often used.

When investigating the status of fatty acids in some groups, it is usual that it group compare with a corresponding control group. Not to take values ​​for FA from other articles or systematic papers. For your work, a suitable control group could have been sedentary men matched for age and BMI. In that case, you could talk about the deficit of certain FAs. This is because the level of essential ALA depends on dietary habits characterized for some areas, and it is not good to compare the values ​​of this FA in Russia with some countries whose eating habits are different. That is why there are no reference values ​​and many more studies on a large number of subjects are needed to declare reference values ​​for FA. Thus, need to analyze the FA profile in healthy sedentary men matched for BMI and age with sportsmen, and compare these groups.

It is not entirely clear why the values ​​for ALA and EPA depending on the level of NO3 or NO2 are shown both in Table 4 and graphically, please explain more precisely.

The authors showed results about correlation analysis of the PP indicators and demonstrated associations between FAs and VO2max/kg and the HR baseline and between the NO2 and HR. However, these results are not discussed. Please, explain why these results are important for your manuscript.

Discussion: Please, precisely describe how EPA attenuates palmitic acid-induced ROS formation (line 298).

In part conclusion, the authors point out that for the first time, the participation of essential n-3PUFAs in the nitrite-nitrate pathway of nitric oxide synthesis in highly trained skiers has been demonstrated. In my opinion, the authors showed only an association between n-3FA and the level of NO in plasma. For participation n-3FA in the nitrite-nitrate pathway of nitric oxide synthesis, needs stronger proof.

Comments on the Quality of English Language

Minor editing of the English language required

Author Response

Dear reviewers and editor!

Thank you and Reviewers for a very extensive and detailed review of our manuscript. We have changed the manuscript based on the reviewers’ comments. Hopefully the manuscript can be published now. Our responses to the comments of the reviewer are presented below.

Sincerely, all authors (Aleksandra Lyudinina,  Olga Parshukova and Evgeny Bojko)

Reviewers 3

Comments for authors.

In this study, the authors investigated the relationships between levels of n-3 polyunsaturated essential fatty acids (n-3 PUFAs) and stable nitric oxide (NO) metabolites in the blood plasma of athletes. These are my suggestions for the manuscript.

- The title is pretentious, so the word " regulate" in the title needs to be changed to "has a relationship with...." or "is associated with .......".

Thank you for your remark. We considered this expert comment.

- Methods: Please describe the method of transesterification of fatty acids with methanol and acetyl chloride. The reference you provided (#13) does not describe this method. It is important because we need to know if you methylated the fatty acids in the whole plasma lipids or only the non-esterified fatty acids (NEFA).

We have taken account of this remark. The plasma profile of fatty acids from total lipids [fraction includes non-esterified FAs, phospholipids, triglycerides and esterified cholesterol] was determined using gas-liquid chromatography. Sample preparation included lipid extraction from plasma and obtaining FA methyl ethers using methanol and acetyl chloride, as described by Lillington et al. (1981) in our modification [Lillington JM, Trafford DJH, Makin HLJ. A rapid and simple method for the esterification of fatty-acids and steroid carboxylic-acids prior to gas-liquid-chromatography. Clinica Chimica Acta 1981;111(1):91-98].

- Line 98: The evaporator temperature was 260°C, did you mean the injector……

We have taken account of this remark.

- Line 100: margarine acid solution (C17:0). Did you mean margaric acid (heptadecanoic acid)

We have taken account of this remark

- Line 103: We have measured the essential n-3 103 ALA, EPA and DHA... Of these fatty acids, only ALA is essential, as it is not synthesized in the body and must be ingested with food. EPA and DHA, however, are synthesized in the body by the enzymatic system of desaturase and elongase and are not essential fatty acids.

In general, we agree with the remark, but according to literature data, all n-3PUFA are classified as essential fatty acids because they must be provided in the diet to initiate the formation of EPA and DHA. The conversion of EPA and DHA from ALA occurs via several reaction steps. However, the complete conversion of ALA to DHA and EPA is less than 1% in and 8 % respectively at LA/ALA ratio of 9-10:1 (Harnack et al., 2009). This inefficient conversion rate is partly attributed to the production of omega-6 PUFA since there is competition for the desaturase and elongase phase and the typical western diet contains a higher intake of omega-6 fatty acids than omega-3 fatty acids (Burdge and Wootton, 2002; Philpott et al., 2018; Shramko et al., 2020).

- Line 110: Please, precisely define NOx. From the sentence, it is not clear what ratio is NOx.

We considered this expert comment.

- It is not clear why the Exercise Test on a Cycle Ergometer "until Exhaustion" was done because all the results presented in the paper refer only to baselines.

Thank you for remark. Data on physical performance in the test to failure were included in order to assess the relationship between biochemical and physiological indicators.

 The statistics chapter is missing, so it is not clear how the results were processed.

We apologize for the absence the statistics chapter, apparently, they didn’t include it after the professional translation

- As for the methods in general, they are written almost identically to the methods in their other papers related to skiers, some of which were also published in the Cells. In my opinion, it needs to be reformulated.

We hope our edits improved the MM section.

Results:

- The abbreviations for FA must be marked the same throughout the work. Namely, in the first part of the paper, you marked alpha-linolenic acid as ALA, and then in the results with α-LNA. Introduce the abbreviations when you first mention these FAs, that is, in the methodology, stating the full name of fatty acids just as you put in Table 2, and the abbreviation for alpha-linolenic acid can be ALA because it is more often used.

Thank you for remark. We introduced abbreviations alpha-linolenic acid as α-LNA throat the all text.

- When investigating the status of fatty acids in some groups, it is usual that it group compare with a corresponding control group. Not to take values for FA from other articles or systematic papers. For your work, a suitable control group could have been sedentary men matched for age and BMI. In that case, you could talk about the deficit of certain FAs. This is because the level of essential ALA depends on dietary habits characterized for some areas, and it is not good to compare the values of this FA in Russia with some countries whose eating habits are different. That is why there are no reference values and many more studies on a large number of subjects are needed to declare reference values for FA. Thus, need to analyze the FA profile in healthy sedentary men matched for BMI and age with sportsmen, and compare these groups.

We considered this expert comment.

In earlier work, we conducted a similar analysis between athletes (cross-country skiers) and young untrained man (Source - Lyudinina A. Yu. Comparative Analysis of the Fatty Acid Profile in the Diet and Blood of Athletes and Students // Human Physiology, 2022, Vol. 48, No. 5, pp. 563568). Due to this remark, we decided to add some of the results of this work to the discussion (line 309).

“Analysis of the fat component of the diet revealed an increased consumption of saturated fats and n-6 PUFAs relative to the recommended norms in both groups. In students, the consumption of essential n-3 eicosapentaenoic (EPA) and docosahexaenoic (DHA) acids was significantly lower compared to skiers (p = 0.013) and the recommended norm. A suboptimal fat diet was accompanied by an imbalance in the FA profile in the blood in both groups. Cross-country skiers have significantly lower levels of saturated myristic (p = 0.000) and palmitic acids (p = 0.003), which are within the reference values. The proportion of essential n-3 linolenic acid in the blood plasma of cross-country skiers is lower than that of students (p = 0.002) and 2.2 times lower than the norm. The level of EPA in the blood in both groups was also reduced, and in students was almost 3 times more pronounced than in skiers (p = 0.000). Thus, the consumption of n-3 PUFAs by athletes in accordance with the recommended norms does not cover their consumption for energy supply and physiological functions involved in intense physical exertion and reduce the aerobic performance of the body”.

- It is not entirely clear why the values ​​for ALA and EPA depending on the level of NO3 or NO2 are shown both in Table 4 and graphically, please explain more precisely.

In addition to n-3  LNA and EPA, the table also presents data on DHA. We would prefer to leave the information both forms in the table and in the figure.

- The authors showed results about correlation analysis of the PP indicators and demonstrated associations between FAs and VO2max/kg and the HR baseline and between the NO2 and HR. However, these results are not discussed. Please, explain why these results are important for your manuscript.

We considered this expert comment.

- Discussion: Please, precisely describe how EPA attenuates palmitic acid-induced ROS formation (line 298).

We hope our edits improved the Discussion section.

- In part conclusion, the authors point out that for the first time, the participation of essential n-3PUFAs in the nitrite-nitrate pathway of nitric oxide synthesis in highly trained skiers has been demonstrated. In my opinion, the authors showed only an association between n-3FA and the level of NO in plasma. For participation n-3FA in the nitrite-nitrate pathway of nitric oxide synthesis, needs stronger proof.

We hope our edits improved the conclusion. “We demonstrated association  essential n-3PUFAs with the level of NO in plasma and possible participation n-3PUFAs in the nitrite-nitrate pathway of nitric oxide synthesis in highly trained skiers has been demonstrated...”

Round 2

Reviewer 1 Report

Comments and Suggestions for Authors

Authors did a good job in revising the manuscript addressing each comments. I still find no answers for the question of why n-6 fatty acids are not investigated? Authors must state the limitations of their method, including a statement about n-6 fatty acids, their quantification method using odd chain fatty acids.

Secondly, the Figure 1 still seems to have missing minimum bar? Could author carefully revise the figure and provide the details in figure legends, including number of participants.

Remove the grid lines in figures. Provide the raw data, peak area, standards vs sample EICs, concentration values as supporting information.

Also, there are discrepancies in the representation of p-value. Unify it. 

Author Response

Reviewers 1

Authors did a good job in revising the manuscript addressing each comments. I still find no answers for the question of why n-6 fatty acids are not investigated? Authors must state the limitations of their method, including a statement about n-6 fatty acids, their quantification method using odd chain fatty acids.

Secondly, the Figure 1 still seems to have missing minimum bar? Could author carefully revise the figure and provide the details in figure legends, including number of participants.

Remove the grid lines in figures. Provide the raw data, peak area, standards vs sample EICs, concentration values as supporting information.

Also, there are discrepancies in the representation of p-value. Unify it.

We did not find any studies on the relationship between the levels of stable metabolites of nitric oxide and n-6 PUFAs among athletes. In the future it will be possible to think in this direction, thank you for the recommendation. We added a section about the limitations of the study, revised the figures and unified the p-values. We hope our edits improved the article.

Reviewer 2 Report

Comments and Suggestions for Authors

The aim was to investigate the relationships between n-3 PUFAs and stable NO metabolites in the blood plasma of athletes. The plasma NO level was study by colorimetric method via reaction with the Griss reagent. The authors claim that: (1) Higher values of EPA promote accumulation of nitrites in the blood of athletes, thus creating conditions for the production of NO; (2) Higher levels of ALA promote an increase in nitrate levels while also regulating NO stores.

Having read the revised version of the manuscript, I remain of the opinion that the work is very weak from a scientific and methodological point of view and does not correspond to the profile of the journal. At the same time, if the editors and other reviewers hold a different point of view, I in no way intend to prevent the publication of this article. Indeed, the authors have made efforts and made a number of corrections in accordance with my comments. The authors' responses to some comments clarified the situation for me, so I am ready to lower the bar of requirements for this article.

At the same time, with a more lenient attitude towards the authors and their manuscript, one cannot accept the fact that a number of corrections were not made, and new semantic or stylistic errors appeared in new fragments. I will list some phrases or wording that require correction; in some places, after this or that phrase there will be explanations in parentheses:

Line 2:   acids is associated the stable nitric oxide metabolites

Line 42-44:  Cross-country skiers are known to have a very high maximal oxygen uptake and are able to perform submaximal exercise at a rather high metabolic rate [4], which may be leads an intensification of the processes of production of reactive oxygen species.

Line 59: these metabolites in the plasma blood

Line 240:  fatty acids in the plasma blood

Line 334-335:  in the plasma of blood of skiers

Line 400-401: nitrite-nitrate (nonenzymatic) pathway of NO synthesis.  (generation, but not synthesis)

Line 403: accumulation of NO    (NO is not accumulated, in this case “generation” should be written)

Line 488: A deficiency of EPA in the plasma of the subjects (0.6±0.5 mol%) was not only related with dietary habits but also may indicate about it participate in neutralization oxidative stress in the body because EPA inhibits the processes of lipid peroxidation in membrane vesicles

Comments on the Quality of English Language

My comments regarding the quality of English are an integral part of the review, examples from the text illustrate this.

Author Response

Reviewers 2

At the same time, with a more lenient attitude towards the authors and their manuscript, one cannot accept the fact that a number of corrections were not made, and new semantic or stylistic errors appeared in new fragments. I will list some phrases or wording that require correction; in some places, after this or that phrase there will be explanations in parentheses:

Line 2:   acids is associated the stable nitric oxide metabolites

Line 42-44:  Cross-country skiers are known to have a very high maximal oxygen uptake and are able to perform submaximal exercise at a rather high metabolic rate [4], which may be leads an intensification of the processes of production of reactive oxygen species.

Line 59: these metabolites in the plasma blood

Line 240:  fatty acids in the plasma blood

Line 334-335:  in the plasma of blood of skiers

Line 400-401: nitrite-nitrate (nonenzymatic) pathway of NO synthesis.  (generation, but not synthesis)

Line 403: accumulation of NO(NO is not accumulated, in this case “generation” should be written)

Line 488: A deficiency of EPA in the plasma of the subjects (0.6±0.5 mol%) was not only related with dietary habits but also may indicate about it participate in neutralization oxidative stress in the body because EPA inhibits the processes of lipid peroxidation in membrane vesicles

We are grateful for Your review with modern understandings of the metabolism of NO and its stable products in the body. We hope our edits improved the article.

Reviewer 3 Report

Comments and Suggestions for Authors

After my suggestion, the manuscript is improved and can be accepted for publication

Author Response

Thank you for your detailed review and positive decision.